

**Short- and long-term temperature responses of soil denitrifier net $N_2O$ efflux rates, inter-**
**profile $N_2O$ dynamics, and microbial genetic potentials**
Buckeridge, Kate M.[1,a,b], Edwards, Kate A.[2], Min, Kyungjin[1,c], Ziegler, Susan E.[3], Billings, Sharon
A.[1]
1.   Department of Ecology and Evolutionary Biology and Kansas Biological Survey,

6       University of Kansas, Lawrence, KS, USA

2.   Canadian Forest Service, Natural Resources Canada, Ottawa, ON, Canada
3.   Department of Earth Sciences, Memorial University, St. John's, NL, Canada
a.   Corresponding author: kmbuckeridge@gmail.com, +44 (0) 1316505093
b.   Present address: Global Academy of Agriculture and Food Security, The Royal (Dick)

11       School of Veterinary Studies, University of Edinburgh, UK

c.   Present address:  Department of Plant and Environmental Sciences, Clemson University,

13       Clemson, SC, USA



**Abstract**

Production and reduction of nitrous oxide ($N_2O$) by soil denitrifiers influences atmospheric concentrations of this potent greenhouse gas. Accurate climate projections of net $N_2O$ flux have three key uncertainties: 1) short- vs. long-term responses to warming; 2) interactions among soil horizons; and 3) temperature responses of different steps in the denitrification pathway. We addressed these uncertainties by sampling soil from a boreal forest climate transect encompassing a 5.2 °C difference in mean annual temperature, and incubating the soil horizons in isolation and together at three ecologically relevant temperatures in conditions that promote denitrification. Both short-term exposure to warmer temperatures and long-term exposure to a warmer climate increased $N_2O$ emissions from organic and mineral soils; an isotopic tracer suggested an increase in $N_2O$ production was more important than a decline in $N_2O$ reduction. Short-term warming promoted reduction of organic horizon-derived $N_2O$ by mineral soil when these horizons were incubated together. The abundance of *nirS* (a precursor gene for $N_2O$ production) was not sensitive to temperature, while that of *nosZ clade I* (a gene for $N_2O$ reduction) decreased with short-term warming in both horizons and was higher from a warmer climate. These results suggest a decoupling of gene abundance and process rates in these soils that differs across horizons and timescales. In spite of these variations, our results suggest a consistent, positive response of denitrifier-mediated, net $N_2O$ efflux rates to temperature across timescales in these boreal forests. Our work also highlights the importance of understanding cross-horizon $N_2O$ fluxes for developing a predictive understanding of net $N_2O$ efflux from soils.

Keywords: nitrous oxide, *nosZ*, *nirS*, boreal forest, [15]N, climate change

Manuscript highlights:

- short- and long-term exposure to warmer temperatures increased soil net $N_2O$ flux
- short-term warming promoted reduction of organic horizon derived $N_2O$ by mineral soil
- gene abundance - process rate coupling in these soils differed across horizons and timescales



## 1. Introduction

Nitrous oxide ($N_2O$) is a potent greenhouse gas, with ~300 times the global warming potential of carbon dioxide on a 100-y timescale and uncertain climate feedback effects (Ciais et al., 2013; Portmann et al., 2012). Though increases in atmospheric $N_2O$ are attributed to N-fertilizer use (Mosier et al., 1998), emissions from natural systems dominate terrestrial fluxes (Ciais et al., 2013) and experimental manipulations indicate warming may enhance these fluxes (Benoit et al., 2015; Billings and Tiemann, 2014; Kurganova and Lopes de Gerenyu, 2010; Szukics et al., 2010; Wang et al., 2014). One of the most important biogeochemical pathways of $N_2O$ formation in natural systems is denitrification, the stepwise reduction of $NO_3^-$ to $N_2$. In this pathway, soil denitrifiers can both produce and reduce $N_2O$, and incomplete reduction of $N_2O$ during the final step to $N_2$ can result in $N_2O$ release to the atmosphere (Baggs, 2011; Firestone and Davidson, 1989). Soil microorganisms play a critical role in climate change (Cavicchioli et al., 2019) yet it remains unclear how sensitive the denitrification pathway is to a warming climate.

Translating empirically-derived knowledge about soil denitrifiers into climate projections is difficult due to the dynamic and variable nature of the many interacting steps and their controls (Butterbach-Bahl et al., 2013). In this study we address three key challenges that are associated with the temperature sensitivity of the emergent process of denitrification. First, we do not know if short-term responses of denitrifying communities to warming (Billings and Tiemann, 2014; Kurganova and Lopes de Gerenyu, 2010; Szukics et al., 2010; Wang et al., 2014) are maintained across longer timescales, and thus we cannot know if laboratory studies can provide the empirical data needed to project longer-term fluxes. Studies of heterotrophic soil $CO_2$ efflux suggest that enhanced rates of microbial respiration with warming may be dampened over the long-term, prompted by a combination of microbial acclimation and adaptation (Billings and Ballantyne, 2013; Bradford, 2013), and it is feasible that denitrifying communities may also exhibit only ephemeral responses to warming. Such a response is consistent with inconclusive results of multiple *in situ* warming experiments though such studies necessarily reflect both denitrification and other $N_2O$-producing processes in soils (Bai et al., 2013; Butler et al., 2012; Dijkstra et al., 2012; McDaniel et al., 2013). Assuming microbial acclimation, soils indigenous to a particular climate regime may harbor denitrifying



communities that are more effective at $NO_3^-$ reduction and transformation to $N_2$ in that
climate's typical temperature range. In principle, this could result in relatively lower rates of
$N_2O$ loss in that particular temperature regime (i.e. more complete denitrification) compared to
less effective processing by those microbial communities if the mean temperature were to shift.
Though this phenomenon has not been demonstrated for the more complicated soil
denitrification with its multiple enzymatic steps, the so-called "home field advantage" has been
demonstrated in studies exploring rates of other soil microbial processes (Alster et al., 2013;
Wallenstein et al., 2013).
A second knowledge gap limiting our ability to project future soil $N_2O$ climate feedbacks is
potential variation with temperature in interactions between microbial production and
reduction of $N_2O$ across soil horizons. Implicit in the concept that such cross-horizon
interactions may control net profile $N_2O$ efflux is the assumption that soil denitrifiers have
different patterns of production and reduction in different horizons. This may arise because the
conditions that control $N_2O$ production or reduction differ between horizons, or it may arise
because the metabolic potentials of the soil microbial community in different horizons are
intrinsically different (Blume et al., 2002; Fierer et al., 2003). Consistent with this idea, Goldberg
and Gebauer (2009) illustrated clear variation in patterns of $\delta^{15}N$ of $N_2O$ across soil depth in
response to drought, which could have been caused by variations in either $N_2O$ production or
reduction (Billings, 2008).  The exchange of substrates between soil horizons thus can be an
important process dictating whole-soil $N_2O$ efflux, and may contribute to apparent
inconsistencies between warming effects in the laboratory and the field (reviewed in Bai et al.
2013).  Indeed, profile interactions have been recently demonstrated as important drivers of
soil $CO_2$ efflux: temperature responses of whole soil core respiration can be distinct from the
sum of those observed for horizons incubated in isolation from each other, likely due to
exchange of substrates and microbes among horizons (Podrebarac et al., 2016).  Though
evidence suggests that $N_2O$ produced in one soil horizon may be reduced in another (Goldberg
and Gebauer 2009), the degree to which this may occur, and why, has not been determined.





A third feature challenging our ability to project soil $N_2O$ effluxes in a warmer climate regime is
the potentially different response to warming of distinct steps in the denitrification pathway
(this may be for one or multiple microbes within the community, that carryout the enzymatic
steps). For instance, if the activity of *nosZ*, a gene that codes for an enzyme catalyzing $N_2O$
reduction, experiences a different response to temperature than *nirK*, a gene coding for an
enzyme catalyzing $NO_2^-$ reduction (and thus $N_2O$ production), the net flux of $N_2O$ may either
increase or decrease with temperature depending on the direction and magnitude of both
responses. Though gene abundances sometimes exhibit decoupling from function (Peterson et
al. 2012), quantifying any changes in these functional gene abundances with temperature can
help discern the propensity for temperature responses of relevant microbial communities'
structure, and thus the driving mechanisms for net $N_2O$ production responses. Differential
responses of these genes' abundances to short-term temperature manipulation have been
observed in grassland soils (an increase in *nosZ* with short-term temperature increases; Billings
and Tiemann, 2014), but it is unknown whether these observations are relevant for soil
microbial communities subjected to long-term exposure to distinct temperature regimes.
In this study, we explore these three issues: short- vs. long-term responses of soil denitrifying
communities' net production of $N_2O$ to warming, the exchange of denitrification-derived $N_2O$
among horizons as a driver of temperature response of net $N_2O$ efflux, and the potentially
different responses of the relative abundances of microbial genes linked to $N_2O$ production vs.
reduction to temperature. We invoked a space for time substitution to test our long-term
warming hypothesis, using a climate transect along which mean annual temperature (MAT)
varies but dominant vegetation, soil type, and soil moisture are similar. To elucidate both short-
and long-term temperature responses of soils' denitrifying communities, we incubated soils
that came from different latitudes and climate regimes along this transect (long-term warming)
for 60 h at 5, 15 and 25 °C (short-term warming), to reflect typical current (5 and 15 °C) and
projected future (25 °C) soil temperatures. Specifically, laboratory incubations of mesic organic
and mineral boreal forest soil horizons were established in conditions that promote
denitrification. To understand the potential for interactions among soil horizons as a driver of
temperature response of net $N_2O$ efflux, we incubated organic and mineral soils both



individually and in combination. We measured net rates of $N_2O$ efflux and abundances of
representative functional genes linked to production and reduction of $N_2O$, and estimated $N_2O$
reduction using an isotopic tracer.
We predicted that short-term warming would enhance net $N_2O$ production in these boreal
soils, as in the majority of past incubation studies (Billings and Tiemann, 2014; Kurganova and
Lopes de Gerenyu, 2010; Szukics et al., 2010; Wang et al., 2014).  As outlined above, we also
tested the hypothesis that a warmer temperature regime over a longer timescale would show
the opposite effect: a dampened net $N_2O$ efflux from the historically warmer soils, where
organic N turnover is faster (Philben et al., 2016), and where denitrifying communities
presumably can function as efficient transformers of $NO_3^-$ to $N_2$ at warmer temperatures
compared to their more northern counterparts. We also hypothesized that $N_2O$ produced in
one horizon would be reduced in the other when incubated together, resulting in lower net $N_2O$
efflux than a simple linear combination of these horizons' individual efflux rates. Specifically, we
anticipated that organic soils, relatively rich in microbial abundance and diversity compared to
mineral soils, would reduce mineral-produced $N_2O$, following dominant diffusion gradients.
Finally, we hypothesized that soils exhibiting higher rates of net $N_2O$ production would exhibit
some combination of increased *nir* abundance and decreased *nos* abundance and associated
higher ratios of *nir:nos* gene abundances, reflecting shifts in microbial genetic potentials with
temperature regime.
**2. Materials and method**
*2.1 Study site and soil sampling*
Soil was collected from three mature forest stands at each of three regions along the
Newfoundland and Labrador Boreal Ecosystem Latitudinal Transect (NL-BELT), Canada (Table 1,
Fig.1; (Ziegler et al., 2017)). NL-BELT spans the north-south extent of the balsam-fir dominated
boreal biome in eastern Canada, from southwest Newfoundland to southeast Labrador. This
transect has long-term (century-scale) temperature regime differences, but otherwise similar
conditions. For instance, the three study regions along this transect (from south to north), the
Grand Codroy, Salmon River, and Eagle River watersheds (Fig. 1), have similar Orthic Humo-





Ferric Podzols (Spodosols; Soil Classification Working Group, 1998) and balsam fir (*Abies*
*balsamea*)-dominated vegetation. The difference in MAT and precipitation is 5.2 °C and 431
mm between Grand Codroy (southern-most) and Eagle River (northern-most) climate stations
(Environment and Climate Change Canada 2108).  The soils are mesic and the regions have an
evaporative demand gradient (Table 1) that considerably reduces the precipitation gradient,
making the transect an excellent proxy for investigating soil temperature responses while
mitigating confounding features of differing soil moisture. Three replicate forest stands were
established in each of the three climate regions, allowing us to assess the influence of long-
term differences in MAT (and associated differences in climate) along the transect without
concerns about pseudoreplication, a rarity in large-scale space-for-time substitutions (Ziegler et
al., 2017)
Two large (30 cm$^2$) peds of organic (LFH or O horizon) and mineral (B horizon) soil were
collected at each forest stand on a different calendar date but an equivalent ecological date:
22-24 October 2013 in Eagle River, 4-5 November 2013 in Salmon River, and 22-23 November
2013 in the Grand Codroy. This pre-freeze, post-growing season period typically exhibits
relatively large and active microbial biomass in northern latitude organic soils (Buckeridge et al.,
2013). The $A_h$ and $A_e$ horizons were not present at all sites so were not included in the
incubation at any site. Each collection was shipped to the University of Kansas (4-5 days transit
in insulated coolers, on ice) and processed immediately.  Because regions were processed as
separate experimental blocks we cannot separate the region and block effects. However, we
confounded these factors knowingly, because we believed ecological date and rapid processing
were more important than minimal differences in laboratory practice between blocks.
*2.2 Incubation and headspace gas collection*
Aboveground vegetation (i.e. moss, herbaceous plants, tree seedlings) was removed from the
peds with scissors. The two peds of organic and mineral soil from each forest site were pooled
within horizon and mixed by hand, producing an organic and mineral sample for each forest.
This process was repeated nine times, for three forests in each of three regions. Subsamples
(fresh mass, organic: 50 g; mineral: 40 g) were placed in half-pint (237 ml) Mason jars. To test



the potential for N₂O producers and reducers from one horizon to interact with their
counterparts in the other horizon, 'combined' samples were also prepared in which an open
container of mineral soil (20 g) was placed within a jar, next to organic soil (25 g) such that they
had a shared headspace but were not physically mixed.  Each sample was replicated for three
temperature incubation scenarios (5, 15 and 25 °C), and three blank jars (no soil) were included
for each temperature. To maximize the potential for denitrification we promoted anaerobic
conditions and substrate diffusion to by evacuating headspace air and replacing with He, and
adjusting water-holding capacity to 80% with a $K^{15}NO_3^-$-N solution ($\delta^{15}N$ 3000 ‰) that added 18
and 1.3 µg N g⁻¹ dw soil to the organic and mineral soil samples, respectively (18x background
levels at the time of sampling, although within the annual range of soil $NO_3^-$ availability based
on unpublished field data). Our approach was distinct from a potential denitrification assay,
which calls for non-limiting C and $NO_3^-$ additions to soils (Pell et al., 1996); instead, we intended
to promote conditions conducive to denitrification using natural C pools and as close to natural
$NO_3^-$ concentrations as was feasible. Therefore, this experiment is not predictive of bulk soil
N₂O rates and instead explores controls on N₂O rates in soil zones with low O₂ concentrations.
Such 'hot spots' for biogeochemical cycles in soils are well-documented (McClain and others

200    2003).

Over 60 h of incubation, we collected headspace gas eight times for determination of N₂O
concentration. The multiple time points verified the robustness of the final 60 h time point
measure and the net result of these samples was used to compare across treatments. The first
sample was collected immediately after initiating the incubations, the second sample was
collected at ~3 hours, and then further samples were collected every ten hours afterwards. At
each collection point 14 ml of headspace gas was removed with a needle and gas-tight syringe
and injected into pre-evacuated 12 ml borosilicate vials with a silicone septum and aluminum
crimp (Teledyne Instruments, Inc., CA, USA); at the second and last collection an additional 14
ml headspace gas was removed and injected into pre-evacuated Exetainers (Labco Ltd., High
Wycombe, UK) for isotopic analysis of N₂O in the headspace. After each gas sampling, He of an
equivalent volume was injected into the incubation vessels to maintain pressure in the



containers.  At the end of the incubation all jars were opened and soils were destructively
harvested to quantify soil inorganic N, and for DNA extraction.
*2.3 $N_2O$ concentration and isotope analysis*
Headspace samples were analyzed for $N_2O$ concentration in an auto-injected 5 ml subsample
on a gas chromatograph fitted with an electron capture detector (CP-3800, Varian), and
calibrated against a four-point standard curve that encompassed the sample range. Blank
corrected headspace concentrations were adjusted for the dilution at each sampling with He
replacement, and rates of net $N_2O$ production were calculated as the average of the 8 sample
collections' rates. Net $N_2O$ flux changed throughout the course of the 60 h incubation; we focus
on the average of these rates to integrate both production and reduction into and aggregate
value across the whole incubation. Samples for isotope analysis ($\delta^{15}N$ of $N_2O$) were submitted
to the University of California, Davis, Stable Isotope Facility, where they were analyzed on a
ThermoFinnigan GasBench + PreCon trace gas concentration system interfaced to a
ThermoScientific Delta V Plus isotope ratio mass spectrometer (Bremen, Germany). Analysis
was conducted with 4 standards of 0.4-10 ppm $N_2O$ in He and a precision of 0.1‰ $^{15}N$.
The change in $^{15}N$ enrichment of the $N_2O$ between incubation sampling times at 3 h and 60h
was used to quantify gross reduction of $N_2O$ to $N_2$ (Billings and Tiemann 2014).  Because our
tracer contained far more $^{15}N$ than is present naturally, any natural fractionation during $N_2O$
reduction was negligible compared to the isotopic signature of the tracer in the $N_2O$ pool, and
we can use $^{15}N_2O$ as a means of assessing $N_2O$ production vs. reduction.  If $^{15}N_2O$ at 60 h is
higher than at 3 h, it suggests the tracer was continuing to flow into the $N_2O$ pool more so than
out of it, and thus that $N_2O$ production outpaced $N_2O$ reduction (transformation into $N_2$) at that
time point. In contrast, if $^{15}N_2O$ at 60 h is lower than at 3 h, it suggests that the tracer was
flowing out of the $N_2O$ pool at a greater pace than it was flowing into it, and thus that $N_2O$
reduction outpaced $N_2O$ production at that time point.  We computed the change in percent of
the $^{15}N$ tracer added that was found in headspace $N_2O$ across incubation time as:
$$Change\ in\ ^{15}N_2O = \left( \left( \frac{^{15}N_2O}{^{15}NO_3^--N\ added} \right) * 100 \right)_{final} - \left( \left( \frac{^{15}N_2O}{^{15}NO_3^--N\ added} \right) * 100 \right)_{initial}$$






where $^{15}N_2O$ is ng of $^{15}N$ in headspace $N_2O$ per g of soil, $^{15}NO_3^--N$ is ng of $^{15}N$ in $NO_3^-$ per g of
soil, final refers to the end of the incubation (~60 h), and initial refers to the first time point at
which change in $^{15}N$ of $N_2O$ was assessed (~3 h).

*2.4 Soil nutrient analysis*
To observe changes in extractable inorganic N during the incubation, we extracted soil
subsamples prior to and following the incubation (fresh mass, organic: 12 g; mineral 10 g) by
shaking for 1 h with 40 ml 0.5 M $K_2SO_4$. After shaking all samples were filtered and extracts
frozen at -20 °C until further analysis. Soil $NO_3^--N$ and $NH_4^+-N$ in the extracts were analyzed on a
Lachat 8500 Autoanalyzer (Hach Co., Loveland, CO, USA) using the cadmium reduction and
phenol red methods, respectively.
*2.5 Functional gene abundance*
Soil DNA was extracted from 0.25 g soil using MoBio Power Soil DNA extraction kit and purified
with MoBio PowerClean DNA Clean-up kit (MoBio Laboratories, Carlsbad, CA, USA, now
Qiagen). DNA was quantified with a Qubit 2.0 Fluorometer (Invitrogen, Carlsbad, CA, USA),
diluted by a factor of ten and stored at -20 °C until further analysis. We assayed several
functional gene primers in the denitrification pathway via PCR, and selected *nirS* (Geets et al.,
2007) and *nosZ clade I* (Wallenstein and Vilgalys, 2005) as the most tractable indicators of $N_2O$
production and reduction in these soils using quantitative PCR (qPCR), based on successful
amplification of these genes across all samples. Note that we were not able to amplify *nirK* or
*nosZ clade II* in these soils. qPCR was accomplished using the ABI StepOnePlus (Applied
Biosystems) with Brilliant III Ultra-Fast SYBR® Green QPCR Master Mix (Agilent/Life
Technologies, Carlsbad, CA, USA). Each reaction consisted of 5 µl (~2 ng) genomic DNA, 400 nM
each primer, 300 nM reference dye and 1 X Brilliant III in a final volume of 20 µl. The qPCR
program consisted of an initial denaturing temperature of 95 °C for 3 min followed by 40 cycles
of denaturing at 95 °C for 5 s and a combined annealing and extension step of 10 s at 60 °C for
both *nirS* and *nosZ* genes. Melt curves were calculated at the end of each qPCR run to confirm



product specificity. Each qPCR plate contained one primer pair, three negative controls and a
four-point standard curve (ranging from 300 to 300,000 copies). Standard curves were
generated using genomic DNA from lab stock of cultured *Pseudomonas fluorescens* and gene
copy numbers were calculated assuming a mass of $1.096 \times 10^{-21}$g per base pair (Wallenstein and
Vilgalys, 2005), one gene copy per genome, and a genome size of 7.07 Mb (NCBI).

*2.6 Statistical analysis*

We used a three-way ANOVA to assess the influence of the fixed effects of soil horizon, 'region'
(historical temperature), 'temperature' (short-term, incubation temperature) and their
interactions on: inorganic N pools, net $N_2O$ flux averaged across the incubation, change in
percent of added $^{15}N$ tracer found in headspace $N_2O$, the effects of mixing horizons in the
incubation on net $N_2O$ flux, and functional gene abundances. For all analyses, we followed up
significant main effects with a Tukey's posthoc analyses and report adjusted *P*-values. For all
variables, we assessed whether they met assumptions required for performing these statistical
tests, and log-transformed variables before analysis when required. All statistical analyses were
performed in R (R Core Team, 2014), using the MASS package (Venables and Ripley, 2003). All
significant ($\alpha = 0.05$) results and interactions are reported except significant main effects when
significant interactions of their terms are reported instead. Errors reported are one standard
error of the mean.

**3. Results**

*3.1 Changes in inorganic N pools after the incubation*

Temperature altered the pool sizes of $NH_4^+$-N differently in each region and horizon (temp x
region x horizon: *P*=0.05), increasing relative to pre-incubation pool sizes in the organic soils at
some of the incubation temperatures (coolest region, 25 °C: *P*=0.04; intermediate region, 25 °C:
*P*=0.02; warmest region, 15 °C: *P*<0.0001, 25 °C: *P*=0.0001) (Fig. 2 A and B). Mineral soil $NH_4^+$-N
pool sizes post-incubation did not differ from pre-incubation pool sizes.
Temperature also altered the pools sizes of $NO_3^-$-N differently for each region and horizon
(temp x region x horizon: *P*=0.03), decreasing relative to pre-incubation pool sizes in the organic



soils at all temperatures in all regions (coolest, 5 ℃: $P=0.001$, 15 ℃: $P=0.0007$, 25 ℃: $P=0.003$;
intermediate, 5 ℃: $P=0.04$, 15 ℃: $P=0.002$, 25 ℃: $P=0.008$; warmest, 5 ℃: $P<0.0001$, 15 ℃:
$P<0.0001$, 25 ℃: $P<0.0001$). $NO_3^--N$ pool sizes also decreased in the mineral soils at all
temperatures in the coolest (5 ℃: $P=0.0005$, 15 ℃: $P=0.0008$, 25 ℃: $P=0.002$) and intermediate
(5 ℃: $P=0.02$, 15 ℃: $P=0.002$, 25 ℃: $P=0.0004$) regions, although not in the warmest region (Fig.
2 C and D). These results imply that the anaerobic conditions we generated by replacing
headspace air with He and keeping 80% water holding capacity generally supported
denitrification and limited nitrification.
*3.2 $N_2O$ net production rates with short- and long-term warming*
Net $N_2O$ flux was influenced by regions ($P=0.002$), incubation temperature ($P=0.006$), and soil
type ($P<0.0001$) without any significant effect of any interaction among or between these
independent variables. When averaged across all incubation temperatures and the two soil
horizons, the warmest region (3.8±0.8 ng $N_2O$-N $g^{-1}$ $h^{-1}$) had a higher rate than the intermediate
(1.9±0.6 ng $N_2O$-N $g^{-1}$ $h^{-1}$, $P=0.008$) and coolest region (1.2±0.3 ng $N_2O$-N $g^{-1}$ $h^{-1}$, $P=0.003$),
whereas the intermediate latitude and coolest regions' net $N_2O$ production did not differ from
each other (Fig. 3). Averaged across all regions and the two soil types, the warmest incubation
temperature (3.4±0.8 ng $N_2O$-N $g^{-1}$ $h^{-1}$) exhibited a higher net $N_2O$ flux than the lowest
temperature (1.1±0.3 ng $N_2O$-N $g^{-1}$ $h^{-1}$, $P=0.003$). Averaged across all regions and soil
temperatures, the organic soil (4.9±0.8 ng $N_2O$-N $g^{-1}$ $h^{-1}$) exhibited a higher rate than the
mineral soil (0.6±0.2 ng $N_2O$-N $g^{-1}$ $h^{-1}$, $P<0.0001$) and the combined incubation (1.3±0.3 ng $N_2O$-
N $g^{-1}$ $h^{-1}$, $P<0.0001$), which had a higher rate than the mineral soil alone ($P=0.005$).

We used $N_2O$ emission from organic and mineral soil in isolation (Fig. 3 A & C) to compute
expected net $N_2O$ flux for the combined soils (Fig. 4 A & B). Observed rates of net $N_2O$
production in the headspace surrounding combined organic and mineral soils (Fig. 3 B) were
less than expected values (Fig. 4 A & B) and often exhibited net $N_2O$ reduction, implying inter-
profile interactions and differential temperature responses of the two horizons. The absolute
effect of the combined horizons' reduction of $N_2O$ differed by incubation temperature



(*P*=0.002), with higher net reduction in the warmest incubation as compared to the coolest (25
vs. 5 °C: *P*=0.001) and a trend towards more reduction in the intermediate latitude region as
compared to the coolest (*P*=0.098). In proportional terms, the effect of combining horizons
decreased the combined net $N_2O$ flux by up to 200% of the expected combined net production
rate, and this effect differed by temperature (*P*=0.009). In particular, it was more pronounced
at 15 °C relative to 5 °C (*P*=0.004). There was no significant interaction between region and
temperature on this combined-horizon rate.
We used the change in $^{15}N$ in the $N_2O$ (t$_{60h}$-t$_{3h}$) as a proxy for estimating how the relative
contribution of production and reduction of $N_2O$ varied among regions, across horizons, and
with incubation temperature. The change in $^{15}N_2O$ across incubation time was consistently
positive, suggesting that rates of $N_2O$ production consistently outpaced rates of $N_2O$ reduction
during the incubation.  These values differed by region (*P*=0.001), a feature driven by the
warmest region exhibiting the largest change compared to the coolest region (*P*=0.0007), and a
similar trend between the warmest and intermediate-latitude regions (*P*=0.081; Fig. 5). There
was no significant effect of incubation temperature or soil type or any interaction between
temperature, region and soil type on this change in $N_2O$-$^{15}N$.

*3.3 Functional gene abundance*

At the end of the 60 h incubation period, the abundance of one functional gene indicative of
$N_2O$ production, *nirS*, did not vary significantly by incubation temperature or region but differed
strongly by soil horizon (*P*<0.0001). There was a higher abundance of this gene in the organic
soil (0.73 x $10^6$ g$^{-1}$ ± 0.04 x $10^6$) vs. the mineral soil (0.18 x $10^6$ g$^{-1}$ ± 0.02 x $10^6$) (Fig. 6).  There
was no significant effect of any interaction among or between the independent variables on
*nirS* abundance. Functional gene abundance for $N_2O$ reduction, *nosZ*, differed by region
(*P*=0.0002), incubation temperature (*P*=0.04) and soil (*P*<0.0001). It was higher in soils from the
warmest region (8.4 x $10^6$ g$^{-1}$ ± 1.9 x $10^6$) relative to the intermediate latitude region (4.0 x $10^6$ g$^{-1}$
± 0.8 x $10^6$, *P*=0.0006) and the coolest region (4.9 x $10^6$ g$^{-1}$ ± 1.1 x $10^6$, *P*=0.001), at the coolest
(6.7 x $10^6$ g$^{-1}$ ± 1.6 x $10^6$) relative to the warmest incubation temperature (5.2 x $10^6$ ± 1.7 x $10^6$,
*P*=0.02), and in organic (10.55 x $10^6$ ± 0.95 x $10^6$) relative to mineral soils (0.98 x $10^6$ ± 0.08 x

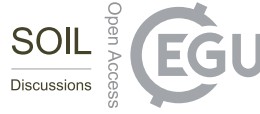

$10^6$). There was no significant effect of any interaction among or between the independent
variables on *nosZ* abundance, although there was a near-significant trend for soil type to alter
the regional effect ($P$=0.052). The resulting *nirS:nosZ* ratio ranged from 0.03 to 0.55 and
displayed an interaction between region and soil horizon ($P$=0.04), driven by lower *nirS:nosZ*
ratios in organic soil in the warmest relative to intermediate latitude region ($P$<0.0001) and
warmest relative to coolest region ($P$=0.003); these effects were not exhibited in the mineral
soil.

**4. Discussion**

By promoting the denitrification pathway we aimed to: 1) distinguish short- (via laboratory
manipulations) and long-term (via a natural climate gradient) responses of denitrification-
derived net $N_2O$ flux to temperature; 2) assess the degree to which net $N_2O$ fluxes in these soils
are sensitive to interactions between soil horizons; and 3) leverage the abundance of genes
responsible for denitrifier production and reduction of $N_2O$ as a means of assessing differences
in these processes' responses to short- and long-term temperature responses. Our first
hypothesis was not supported: though short-term warming enhanced net $N_2O$ effluxes from
these soils, soils from a historically warmer environment exhibited greater net $N_2O$ efflux than
those from cooler environments, suggesting a positive response of net $N_2O$ fluxes to both short-
and long-term warming (Fig. 3). Indeed, an isotopic proxy for $N_2O$ reduction derived from use of
a stable isotope tracer suggests that enhancement of net $N_2O$ production with long-term
warming is greater than any enhancement in $N_2O$ reduction (Fig. 5). Our second hypothesis was
supported in that the combined incubation of mineral and organic soils exhibited net $N_2O$ efflux
rates that did not match the linear sum of separate incubation flux rates. However, we
observed reduction of $N_2O$ by mineral soil, not by organic soil as we predicted. Specifically, net
$N_2O$ production was tempered by more mineral soil $N_2O$ reduction at warmer incubation
temperatures (Fig. 4 & 5), indicating that soil horizon interactions may be critical to rates of net
$N_2O$ efflux to the aboveground atmosphere. Finally, our third hypothesis that linked gene
abundance to process rates was only partially supported. *NosZ* decreased at the warmest
incubation temperature (i.e. lower $N_2O$ reduction gene abundance with warming, Fig. 6),



consistent with rates.  However, in the organic soils, *nosZ* was higher under higher historical
temperature (i.e. higher $N_2O$ reduction gene abundance with warming, Fig. 6), inconsistent with
rates that increase with warming. There was no response to either short- or long-term warming
in *nirS* abundance in either soil horizon, or to long-term warming in *nosZ* abundance in the
mineral soil. Combined, these data suggest complex microbial responses to short- and long-
term exposure to distinct temperature regimes, which we expand upon below.
*4.1 Warming-induced enhancement of $N_2O$ production exceeds that of $N_2O$ reduction*
Long-term climate gradients substitute space for time and encompass variation in multiple
ecosystem phenomena driven by centuries of exposure to distinct climate regimes. For
instance, we know that *in situ* soil N cycling is more rapid (Philben et al., 2016) and likely
supports greater forest productivity in the relatively warm, southern-most boreal forests of this
transect (Ziegler et al., 2017). The net $N_2O$ efflux rate data from this set of lab incubations
suggests that, especially in the organic soil horizons, both short-term warming and a long-term
warmer climate enhance net $N_2O$ production, a result consistent with the stable isotope tracer
data (Fig. 5).  These data correspond with the enhanced, short-term warming-induced $N_2O$
fluxes observed in several systems (Billings and Tiemann, 2014; Kurganova and Lopes de
Gerenyu, 2010; Szukics et al., 2010; Wang et al., 2014).  The apparent lack of long-term,
denitrifier adaptation to rising temperatures (i.e. continued enhancement of $N_2O$ production
with long-term exposure to warmer temperatures that outstrips enhancement of $N_2O$
reduction) is consistent with recent work in soils from these same sites demonstrating no
change in the responses of microbial biomass-specific decay or $CO_2$ efflux rates to warmer
temperatures over decadal timescales (Min et al., 2019).  However, results from the current
study contrast with our predictions of microbial adaptations to a warmer climate over the long
term, which assume that a soil denitrifying community well-adapted to its temperature regime
is adept at complete denitrification with relatively little $N_2O$ byproduct. Such predictions arise
from more conceptual studies presenting ideas about microbial metabolic responses to
warming (Billings and Ballantyne, 2013; Bradford, 2013).

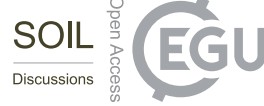

The similar difference in net $N_2O$ rates between the northern region and southern region (2.6
ng $N_2O$-N $g^{-1}$ $h^{-1}$) and between the coolest and warmest incubation temperature (2.3 ng $N_2O$-N
$g^{-1}$ $h^{-1}$, both 68% of the average range across treatments) indicates that net rates were
enhanced to a similar degree by both short-term warming of 20 °C and a long-term MAT
difference of 5 °C. Temperature sensitivity (i.e. change per °C) of net $N_2O$ flux increased at
lower latitudes, and the isotopic tracer experiment indicated that $N_2O$ production increases
outpaced $N_2O$ reduction increases in warmer regions.  Enhanced soil organic matter inputs and
nitrogen availability and cycling rates in the warmer climate forests (Philben et al., 2016; Ziegler
et al., 2017) may contribute to greater net $N_2O$ production. The additive, positive result from
both historically warmer soils and warmer incubation temperatures suggests that community-
level denitrifier performance declines (i.e. more incomplete denitrification) in warmer
temperatures if they are from soils with historically warmer temperatures. This pattern
contradicts a "home-field" advantage (Wallenstein et al., 2013) for denitrifiers.  More $N_2O$
production in warmer climates may arise from multiple changes that overcome adaptive home-
field advantages, such as shifts in the community composition (Delgado-Baquerizo et al., 2016)
and an increased number of inefficient $N_2O$ producers, increases in the number of microbial
cells and transfer points involved in the denitrification pathway (i.e. nitrifier-denitrification in a
single organsism vs. coupled nitrification-denitrification in distinct organisms (Butterbach-Bahl
et al., 2013), or a changed contribution of alternate, possibly less-efficient electron donors (i.e.
co-denitrification (Spott et al., 2011)).
Despite increased net $N_2O$ production to temperature, soil horizon interactions temper the
response to warming. Two of our methods either supported or did not contradict the potential
for mineral soil $N_2O$ reduction: (1) calculated differences in flux values between shared
headspace $N_2O$ flux values and the isolated headspace $N_2O$ flux values of the two isolated
horizons, and (2) the change in isotopic enrichment of the shared and isolated headspace $N_2O$.
The first method demonstrated that short-term warming enhanced the degree of interprofile
interaction that increased $N_2O$ reduction during the incubation, while long-term warming did
not significantly influence interprofile $N_2O$ dynamics (Fig. 4 A & B). The similarities in net $N_2O$



flux between the combined and mineral soil incubations (Fig. 3 B & C) indicate that the mineral

soil served as a net $N_2O$ reducer, especially in response to short-term temperature increases.

Our second method of detecting horizon interactions driving net $N_2O$ efflux used $^{15}N_2O$

headspace differences from the start to the end of the incubation as an indicator of reduction.

We expected an increase in the $^{15}N$ in the headspace $N_2O$ as $^{15}NO_3^-$ is reduced, followed by a

decline in $^{15}N$ in the headspace $N_2O$ as the tracer flows into the $N_2$ pool, with balance of these

processes indicating net production or reduction (Billings and Tiemann, 2014). $NO_3^-$ pools

declined and the change in our $^{15}N_2O$ was positive, suggesting that $N_2O$ production still

outweighed reduction at the end of the 60 h for both the individual horizons and the

combination incubation (Fig. 5 A). Large variation in $^{15}N_2O$ changes among forest sites led to no

significant difference between soil horizons and did not allow us to confirm our horizon

interactions, although these results do not contradict the possibility of mineral soil reduction.

Horizon interactions drove net profile $N_2O$ fluxes in a field drought manipulation in a Norwegian

spruce forest, during which soils exhibited a net $N_2O$ sink via upper mineral soil reduction of

deep mineral soil $N_2O$ production (Goldberg and Gebauer, 2009). It remains unknown if the

relatively shallow mineral soils we sampled are analogous reducers of deeper mineral soil $N_2O$

produced in this system, or if they could continue to reduce large portions of organic soil $N_2O$

efflux (Fig. 4) in situ.

Mineral soil reduction of organic soil-generated $N_2O$ becomes most relevant when diffusion of

$N_2O$ from the upper soil profile to the atmosphere is restricted, and $N_2O$ produced in those

surface layers diffuses downwards according to Fick's Law as has been discussed in the

literature for soil $CO_2$ dynamics (Oh et al., 2005; Richter et al., 2015). Such a situation is likely to

occur in 'hot spots' (McClain et al., 2003) such as frozen surface soil patches during winter.

Similarly, 'hot moments' may occur in the spring snow melt or in winter, despite cold

temperatures reducing N cycling rates: subnivial $N_2O$ production can be an important

contribution to annual N budgets in pastures (reviewed in Uchida and Clough 2015), and winter

N dynamics also appear to be important in northern temperate forest systems.  For example,

winter $N_2O$ production equaled  ~30% of the summer $N_2O$ production in a SE Canadian forest



(Enanga et al., 2016) and ~60% of the annual atmospheric N inputs in a NE U.S. forest (Morse et
al., 2015). Mineral soil reduction of winter organic soil-generated $N_2O$ may temper net fluxes
and may be an important feature in forest N cycling.
*4.2 Linking biogeochemical process rates to genetic potential*
The functional gene that we could quantify in these soils that is associated with $N_2O$ reduction
was sensitive to both short-term and historical temperature, though it was not consistently
associated with process rates. Although we did not detect the atypical *nosZ clade II* in these
soils, other, yet unknown genes that we did not measure may be responsible for $N_2O$ reduction.
Beyond this possibility, our results suggest a decoupling of process rates and denitrifier genetic
controls, or that the long-term temperature-related increase in genetic potential for $N_2O$
reduction did not translate to rates as effectively as the short-term temperature-related
decrease in genetic potential for $N_2O$ reduction.
Consistent with enhanced net $N_2O$ production in these soils at warmer incubation
temperatures, the *nosZ* abundances were reduced after 60 h exposure to 25°C relative to
cooler incubations. Although functional gene abundances are assumed to integrate longer-term
changes in the microbial community and thus have a reduced dynamism relative to
instantaneous rates (Petersen et al., 2012), our results appear to reflect a capacity of
denitrifiers to respond rapidly to temperature, as indicated in other laboratory incubations that
assayed temperature responses of denitrification functional gene abundances (Billings and
Tiemann, 2014; Cui et al., 2016; Keil et al., 2015). However, inconsistent with enhanced net $N_2O$
production in the soils from warmer historical temperatures, we found a reduced *nirS:nosZ*
*Clade I* ratio in the southern forest soils. A possible explanation of this apparent decoupling
between gene abundances and biogeochemical outcomes may be an interference between
potential and transcription (i.e. better detected with mRNA), or inadequate measurement of all
genes relevant to $N_2O$ dynamics in these soils. Although our experimental set up promoted
denitrification, our incubation may have also supported dissimilatory nitrate reduction to
ammonium (DNRA,(Schmidt et al., 2011)). This pathway is poorly characterized, but has been
detected in both aerobic and anaerobic environments of many soil types; it may account for a



large proportion of $NO_3^-$-N reduction in forest soils (Bengtsson and Bergwall 2000). DNRA
represents a process that can reduce $NO_3^-$ via a different nitrite reduction enzyme (*nrf*) than
denitrification (*nir*) and can result in an accumulation of $NH_4$-N, as we observed during our
incubation.  The process also produces and reduces $N_2O$ (Luckmann et al., 2014). The potential
existence of this alternate pathway of $NO_3^-$ reduction and $N_2O$ production and reduction does
not negate the observed $N_2O$ efflux or *nosZ* response to short-term and historical temperature
shifts; however, it does imply that a deeper understanding of the complex genetic N-cycle is
required to link soil process rates to genetic potential.

Contrasting efficiencies of $N_2O$ scavenging is another possible explanation for the decoupling
between gene abundances and biogeochemical fluxes in these soils.  The observation that
mineral soil has the capacity to reduce a substantial amount of organic soil-derived $N_2O$ even as
*nosZ* abundances are reduced in mineral compared to organic soil provides a strong indication
that *nosZ* in mineral soil is more efficient at scavenging $N_2O$ from the headspace than *nosZ* in
the organic horizon.  Consistent with our combination samples in the current study, there is
increasing evidence that soils can serve as sinks for atmospheric $N_2O$ (Chapuis-Lardy et al.
2007), and interestingly, that this phenomenon can be particularly evident when soil water is
limited (Goldberg and Gebauer, 2009). Therefore, given the varying gene abundance and
enzyme efficiency with depth implied in this study, a likely fruitful area of research would be to
explore mineral soil $N_2O$ sink capacity and mineral soil genetic response as moisture availability
varies, as happens particularly during snowmelt periods and in fall within these boreal soils.

**5. Conclusions**
The sensitivity of soil $N_2O$ efflux to global change factors such as temperature can be high, as
supported by this study, but the mechanisms driving $N_2O$ sources and sinks remain challenging
to elucidate.  Indeed, variation of net soil denitrifier $N_2O$ efflux within climate region in this
study, though less than variation across regions, warrants further consideration of within-
region controls on $N_2O$ efflux.  The meaningful across-climate region responses we observed,
though, permitted us to address the three critical issues framed at the outset of this study; we



conclude with three observations and questions for future research. To improve Earth system
models of greenhouse gas emissions we need to address the importance of varying $N_2O$
dynamics with soil depth.  Indeed, this research highlights potentially different efficiencies of
$N_2O$-relevant functional genes as we move across depth.  Is it ubiquitous that *nosZ* is more
efficient in sub-surface soils?  We have taken the first step towards this characterization, but
similar studies should address this question in diverse ecosystems. Our results also illustrate
that both denitrifier-mediated rates of $N_2O$ production and reduction can increase with
warming, over both short- and long-term timescales, in boreal forest soils. *In situ* variables
would undoubtedly alter the *ex situ* fluxes observed in this study, but we demonstrate that
when conditions promote denitrification, the net response to warming in these boreal forest
soils is dominated by $N_2O$ production. Finally, we remain uncertain of the relative importance of
the denitrification pathway in $N_2O$ emissions in boreal forest soils (i.e. as compared to
nitrification, co-denitrification, DNRA and others) and suggest similar approaches to explore the
importance of historic climate regime and interactive responses among soil horizons in other
biochemical pathways of soil $N_2O$ emission.

**Acknowledgements**
We gratefully acknowledge field assistance from Andrea Skinner, and laboratory assistance
from Carl Heroneme, Samantha Elledge, Yanjun Chen and Mitch Sellers. Research funding was
provided by the National Science Foundation (NSF-DEB 0950095) to SAB, Natural Sciences and
Engineering Research Council of Canada (RGPIN#341863) to SZ, an Association for Women
Geoscientists Graduate Research Scholarship and the University of Kansas, and the Kansas
Biological Survey Graduate Summer Research Fund to KM.  The Canadian Forest Service of
Natural Resources Canada provided valuable logistical support.

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



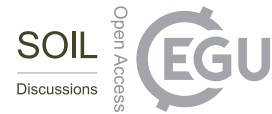



**Table 1.** Characteristics of the nine forests in the three study regions in NL-BELT.

| Region | Coolest | | | Mid | | | Warmest | | |
|---|---|---|---|---|---|---|---|---|---|
| Forest ID | Muddy Pond | Sheppard's Ridge | Harry's Pond | Hare Bay | Tuckamore | Catch-A-Feeder | O'Regans | Maple Ridge | Slug Hill |
| Latitude | 53°33'N | 53°33'N | 53°35'N | 51°15'N | 51°9'N | 51°5'N | 47°53'N | 48°0'N | 48°0'N |
| Longitude | 56°59'W | 56°56'W | 56°53'W | 56°8'W | 56°0'W | 56°12'W | 59°10'W | 58°55'W | 58°54'W |
| Watershed | Eagle River | | | Salmon River | | | Grand Codroy | | |
| Closest weather station ∞ | Cartwright (53°42'N, 57°02'W) | | | Main Brook (51°11'N, 56°01'W) | | | Doyles (47°51'N, 59°15'W) | | |
| Mean annual precipitation (mm) | 1073.5 | | | 1223.9 | | | 1504.6 | | |
| MA PET (mm) ¶ | 432.9 | | | 489.1 | | | 608.1 | | |
| Mean annual temperature (°C) | 0.0 | | | 2.0 | | | 5.2 | | |
| Organic horizon depth (cm) | 6.5 | 4.6 | 6.1 | 9.4 | 7.4 | 6.6 | 7.9 | 8.8 | 4.3 |
| Bulk density (organic) (g cm$^{-3}$) | 0.09 | 0.07 | 0.10 | 0.09 | 0.09 | 0.12 | 0.09 | 0.14 | 0.10 |
| Bulk density (mineral) (g cm$^{-3}$) | 0.80 | 0.72 | 0.76 | 0.59 | 0.59 | 1.20 | 0.68 | 0.68 | 0.66 |
| Soil pH (organic) | 5.3 | 5.3 | 5.4 | 4.4 | 4.4 | 5.7 | 4.3 | 3.7 | 4.6 |
| Soil pH (mineral) | 5.0 | 5.0 | 5.0 | 4.8 | 4.8 | 5.9 | 4.5 | 4.7 | 4.9 |

∞ Climate normal data (1981 - 2000) (http://climate.weather.gc.ca/climate_normals/index_e.html)

¶ MA PET, mean annual potential evapotranspiration




**Figure legends**

Figure 1. a) Map and b) pictures of the three forests in each region along the Newfoundland and Labrador Boreal Ecosystem Latitude Transect in Canada.

Figure 2. Soil $NH_4^+$-N and $NO_3^-$-N pools in the organic (A and C) and mineral soil (B and D), pre-incubation ('Pre-inc.') and at the end of the incubations at 5, 15, and 25°C of soils from along a boreal forest latitudinal transect. Pre-incubation values for nitrate are calculated as ambient concentrations plus added $NO_3^-$-N. Note different y-axis values. 'MAT' = mean annual temperature; the 'coolest' region is the Eagle River watershed (northern boreal), the 'intermediate' region is the Salmon River watershed (mid-boreal), and the 'warmest' region is the Grand Codroy watershed (southern boreal). See text for description of sites. Values provided as the mean ± one standard error (n=3 forests per latitudinal region).

Figure 3. Net $N_2O$ flux ('production rate') averaged for 60 h of incubation at 5, 15, and 25°C from organic soil alone (A), combined organic and mineral soil (B) and mineral soil alone (C) from three regions along a boreal forest latitudinal transect. 'Combined' refers to incubations with organic and mineral soil in the same jar, physically isolated but with shared headspace. 'MAT' = mean annual temperature; the 'coolest' region is the Eagle River watershed (northern boreal), the 'intermediate' region is the Salmon River watershed (mid-boreal), and the 'warmest' region is the Grand Codroy watershed (southern boreal). See text for description of sites. Values provided as the mean ± one standard error (n=3 forests per latitudinal region).

Figure 4. The combination effect of shared headspace surrounding physically separated organic and mineral horizons on the absolute net $N_2O$ flux (A) and as a percent of the expected $N_2O$ production rate (B), at the end a 60 h incubation at 5, 15, and 25°C, for soils from three regions along a boreal forest latitudinal transect. The combination effect (negative = reduction) is calculated as the difference between observed net $N_2O$ fluxes when soil horizons shared the incubation headspace (observed) and the linear, additive effect of rate differences between horizons in separate headspaces (expected).  The non-zero values suggest that the shared headspace generated a non-linear, interactive effect on net $N_2O$ effluxes.  'MAT' = mean annual





temperature; the 'coolest' region is the Eagle River watershed (northern boreal), the

'intermediate' region is the Salmon River watershed (mid-boreal), and the 'warmest' region is

the Grand Codroy watershed (southern boreal).  See text for description of sites. Values

provided as the mean ± one standard error (n=3 forests per latitudinal region).

Figure 5. Change in the % of added $^{15}N$ observed in headspace $N_2O$ over the course of a 60 h

incubation at 5, 15, and 25°C ($t_{60h}$ – $t_{3h}$) for organic (A), combined organic and mineral (B) and

mineral (B) soils from three regions along a boreal forest latitudinal transect. 'Combined' refers

to incubations with organic and mineral soil in the same jar, physically isolated but with shared

headspace. 'MAT' = mean annual temperature; the 'coolest' region is the Eagle River watershed

(northern boreal), the 'intermediate' region is the Salmon River watershed (mid-boreal), and

the 'warmest' region is the Grand Codroy watershed (southern boreal).  See text for description

of sites. Values provided as the mean ± one standard error (n=3 forests per latitudinal region).

Figure 6. Functional gene abundances during a 60-hr incubation at 5, 15, and 25°C from soil

from three boreal forest regions along a latitudinal transect: *nirS* in the organic (A) and mineral

(B) soil; *nosZ* in the organic (C) and mineral (D) soil; and the ratio of *nirS*:*nosZ* in the organic (E)

and mineral (F) soil. Note y-axis scales differ for each row, and between (C) and (D). 'MAT' =

mean annual temperature; the 'coolest' region is the Eagle River watershed (northern boreal),

the 'intermediate' region is the Salmon River watershed (mid-boreal), and the 'warmest' region

is the Grand Codroy watershed (southern boreal).  See text for description of sites. Values

provided as the mean ± one standard error (n=3 forests per latitudinal region).



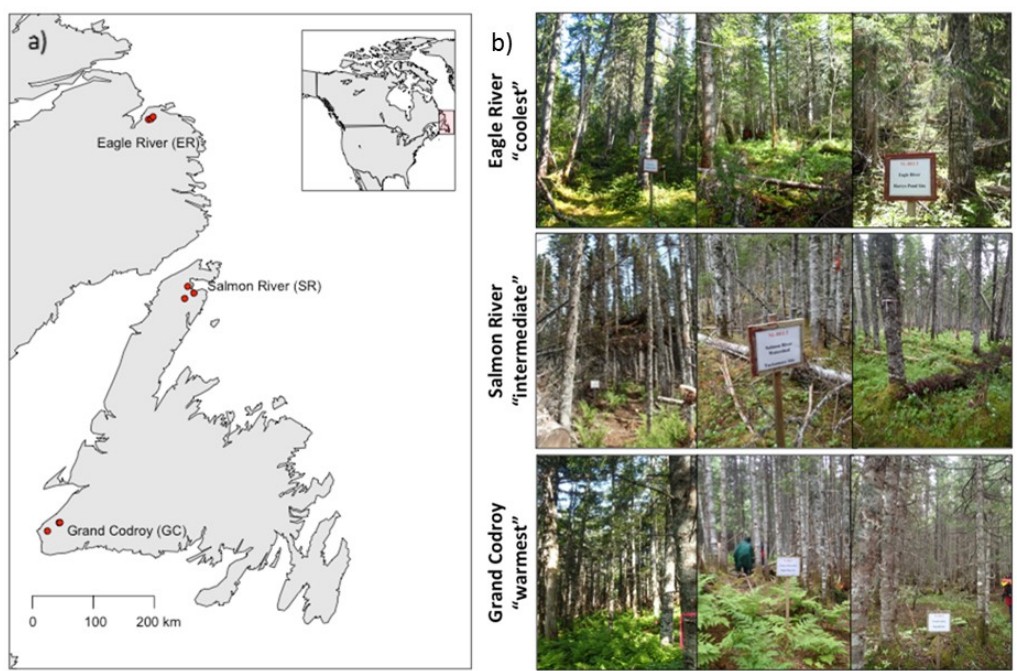


**Figure 1**. a) Map and b) pictures of the three forests in each region along the Newfoundland
and Labrador Boreal Ecosystem Latitude Transect in Canada.



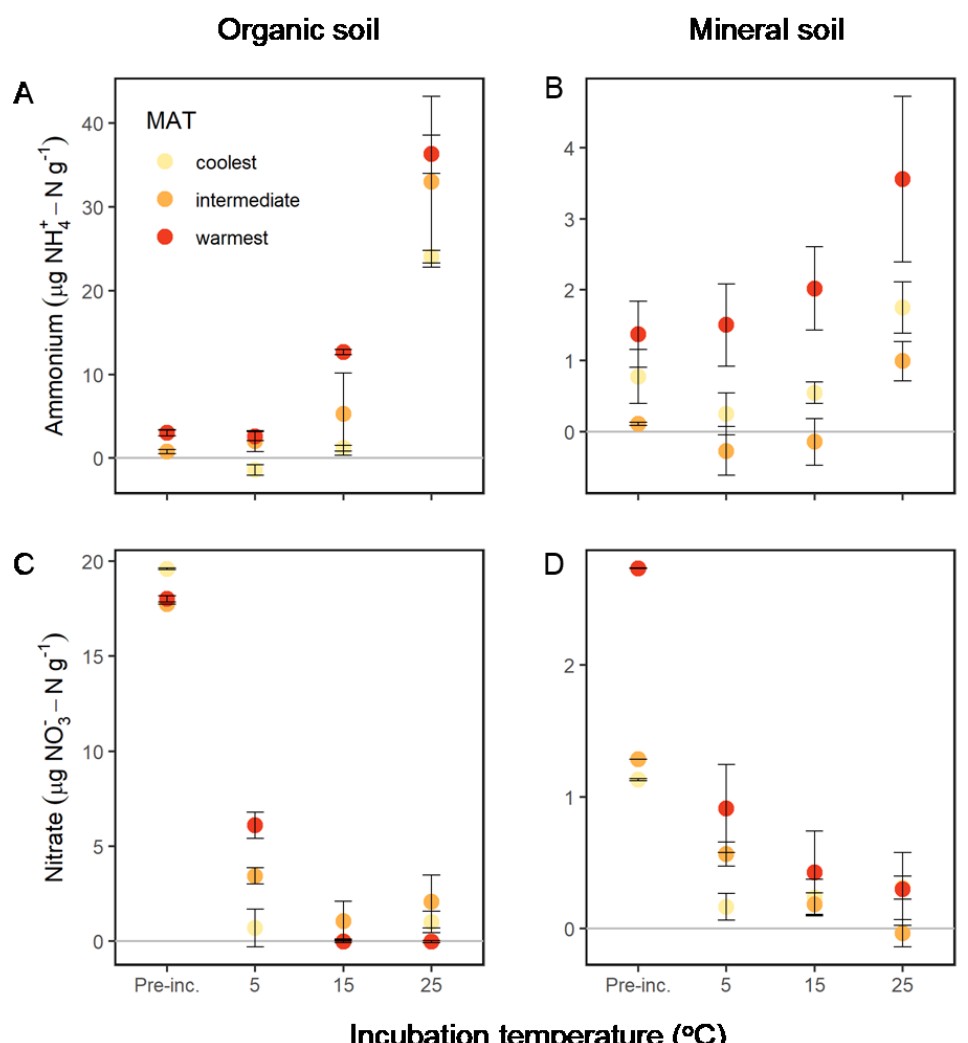

**Figure 2**. Soil NH$_4^+$-N and NO$_3^-$-N pools in the organic (A and C) and mineral soil (B and D), pre-incubation ('Pre-inc.') and at the end of the incubations at 5, 15, and 25°C of soils from along a boreal forest latitudinal transect. Pre-incubation values for nitrate are calculated as ambient concentrations plus added NO$_3^-$-N. Note different y-axis values. 'MAT' = mean annual temperature; the 'coolest' region is the Eagle River watershed (northern boreal), the 'intermediate' region is the Salmon River watershed (mid-boreal), and the 'warmest' region is the Grand Codroy watershed (southern boreal). See text for description of sites. Values provided as the mean ± one standard error (n=3 forests per latitudinal region).



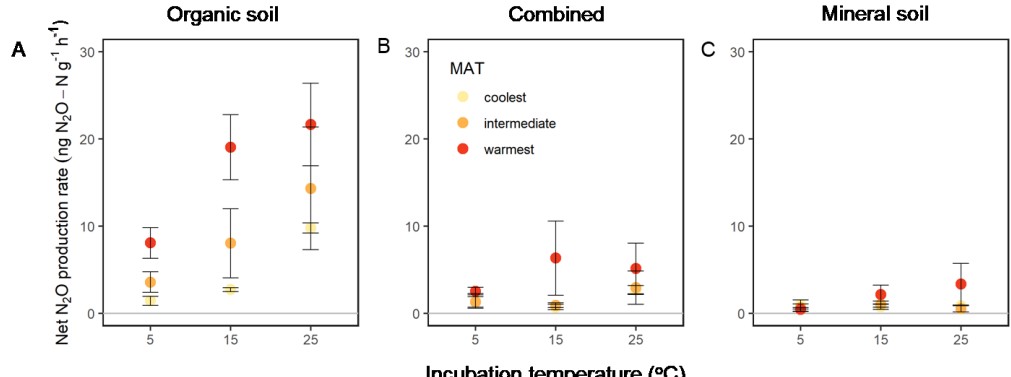

765

**Figure 3**. Net $N_2O$ flux ('production rate') averaged for 60 h of incubation at 5, 15, and 25°C
from organic soil alone (A), combined organic and mineral soil (B) and mineral soil alone (C)
from three regions along a boreal forest latitudinal transect. 'Combined' refers to incubations
with organic and mineral soil in the same jar, physically isolated but with shared headspace.
'MAT' = mean annual temperature; the 'coolest' region is the Eagle River watershed (northern
boreal), the 'intermediate' region is the Salmon River watershed (mid-boreal), and the
'warmest' region is the Grand Codroy watershed (southern boreal). See text for description of
sites. Values provided as the mean ± one standard error (n=3 forests per latitudinal region).

774



775

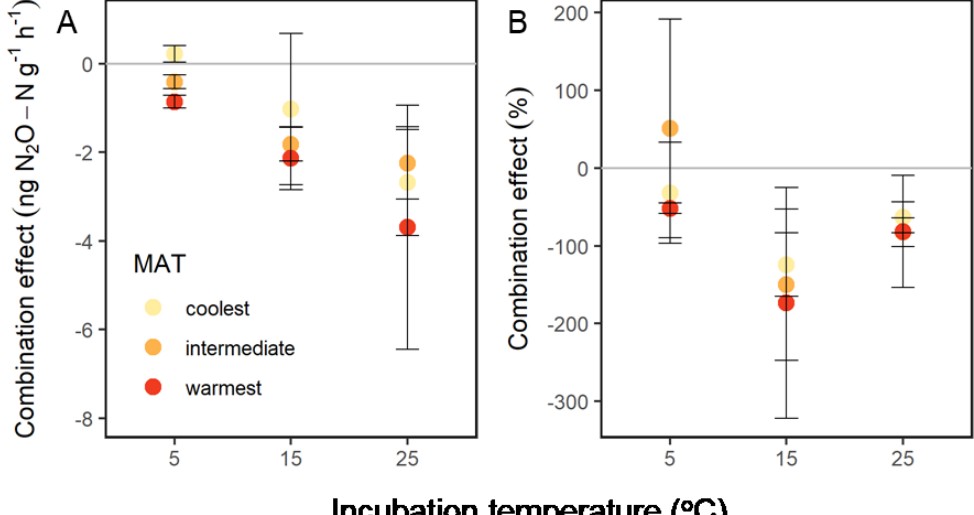

776

**Figure 4**. The combination effect of shared headspace surrounding physically separated organic
and mineral horizons on the absolute net $N_2O$ flux (A) and as a percent of the expected $N_2O$
production rate (B), at the end a 60 h incubation at 5, 15, and 25°C, for soils from three regions
along a boreal forest latitudinal transect. The combination effect (negative = reduction) is
calculated as the difference between observed net $N_2O$ fluxes when soil horizons shared the
incubation headspace (observed) and the linear, additive effect of rate differences between
horizons in separate headspaces (expected).  The non-zero values suggest that the shared
headspace generated a non-linear, interactive effect on net $N_2O$ effluxes.  'MAT' = mean annual
temperature; the 'coolest' region is the Eagle River watershed (northern boreal), the
'intermediate' region is the Salmon River watershed (mid-boreal), and the 'warmest' region is
the Grand Codroy watershed (southern boreal).  See text for description of sites. Values
provided as the mean ± one standard error (n=3 forests per latitudinal region).






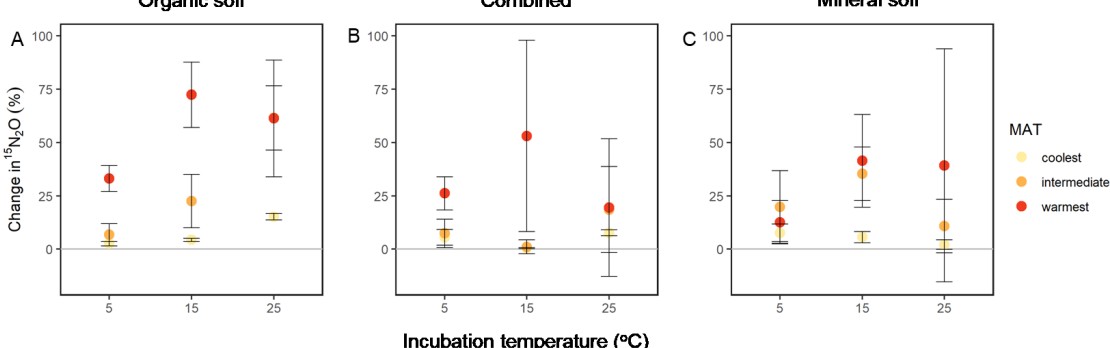


**Figure 5**. Change in the % of added [15]N observed in headspace $N_2O$ over the course of a 60 h
incubation at 5, 15, and 25°C ($t_{60h} - t_{3h}$) for organic (A), combined organic and mineral (B) and
mineral (B) soils from three regions along a boreal forest latitudinal transect. 'Combined' refers
to incubations with organic and mineral soil in the same jar, physically isolated but with shared
headspace. 'MAT' = mean annual temperature; the 'coolest' region is the Eagle River watershed
(northern boreal), the 'intermediate' region is the Salmon River watershed (mid-boreal), and
the 'warmest' region is the Grand Codroy watershed (southern boreal).  See text for description
of sites. Values provided as the mean ± one standard error (n=3 forests per latitudinal region).

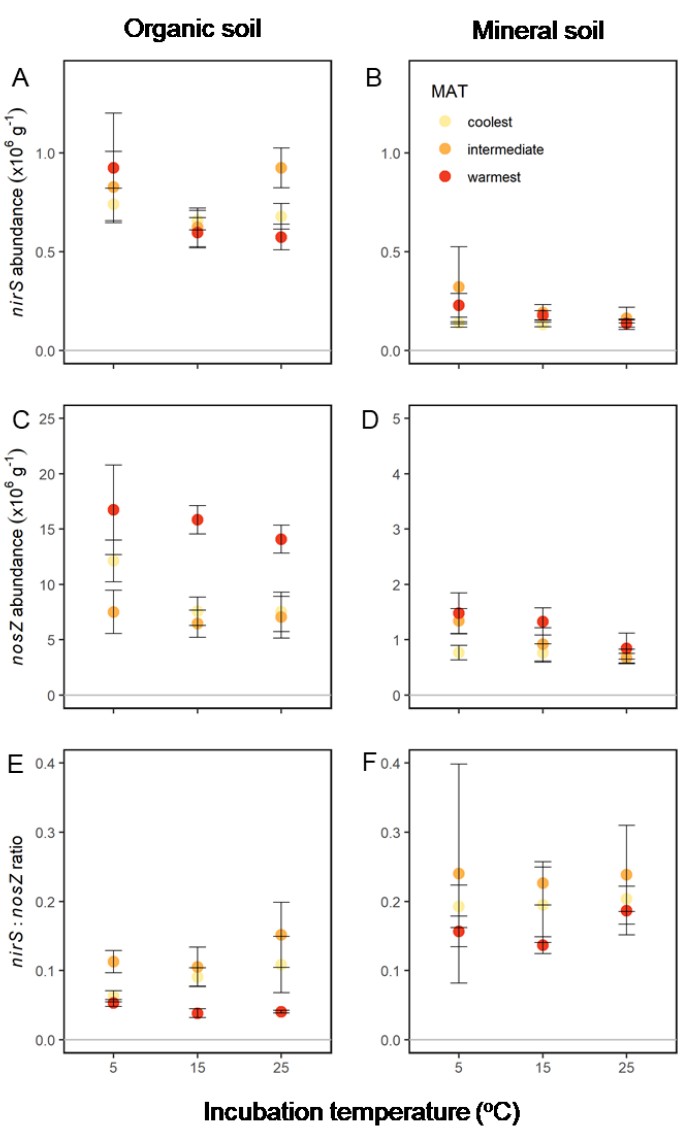


**Figure 6**. Functional gene abundances during a 60-hr incubation at 5, 15, and 25°C from soil
from three boreal forest regions along a latitudinal transect: *nirS* in the organic (A) and  mineral
(B) soil; *nosZ* in the organic (C) and mineral (D) soil; and the ratio of *nirS*:*nosZ* in the organic (E)
and mineral (F) soil. Note y-axis scales differ for each row, and between (C) and (D). 'MAT' =
mean annual temperature; the 'coolest' region is the Eagle River watershed (northern boreal),
the 'intermediate' region is the Salmon River watershed (mid-boreal), and the 'warmest' region
is the Grand Codroy watershed (southern boreal).  See text for description of sites. Values
provided as the mean ± one standard error (n=3 forests per latitudinal region).