# Peer review of "Short- and long-term temperature responses of soil denitrifier net $N_2O$ efflux rates, inter- profile $N_2O$ dynamics, and microbial genetic potentials"

_SOIL, 2019_

## Referee Comment (RC1) · Anonymous Referee #1 · 24 Apr 2020

Overall, this is a very interesting manuscript that is presented in a clear and easy to follow manner. The objectives and hypotheses are presented well and are followed with an exceptional design and conclusions. I thought the use of the combined incubations was a novel approach to provide a link to landscape-scale processes from the experimental setup. The site selections provide a very useful

The manuscript presents very interesting results especially regarding the potential for N2O reduction in mineral soils, but the impact of this finding is somewhat muddled in the presentation of the data.

[Figure]

In figure 4, the combitionation effect is presented and presented as a percent of the 'expected N2O production rate', but the definition of this rate is not clear. The logic behind these calculations is not clearly explained in the manuscript itself. Is the expected rate actually the rate at the end of the 60h incubation of the soils incubated in isolation? So that the values expressed in Fig 4B are the absolute rates of Fig4A divided by the rates in Fig3 A &C? It would be important for the reader to understand the logic behind figure 4 and clearly explain the calculations within the manuscript.

Additionally, the results from the incubations of mineral soil horizons demonstrate low rates of N2O production, but without confirmation of 15N-N2 measurements, how are the authors confident that these low production rates correlate to high N2O reduction rates? This can be addressed with the rate of 15N-NO3 throughout the experiment, but it is not clear in the text as the manuscript is currently written, please elucidate on this in the discussion.

Finally, one very minor comment regarding the figures. To help distinguish between the incubated soils, please use symbols additional to the colors.

---

## Referee Comment (RC2) · Anonymous Referee #2 · 30 Apr 2020

The study is important, since emissions from terrestrial systems dominate N2O fluxes which may be further enhanced by warming climate leading also to warming of the soil.

In this study, the key studied function was denitrification, both producing and consuming N2O. The N2O flux potentials were measured from boreal ecosystem soils where mean annual temperature spans from 0 - 5.2 C. Furthermore, incubations by adding 15N-KNO3 were done in temperature range from 5 to 25 C in order to see the effect of warming to fluxes.

Both short-term warming and a long-term warmer climate enhance net N2O production, and N2O production was bigger than N2O reduction during the incubation. There was reduction of N2O by mineral soil, not by organic soil. Combining horizons of mineral soil and organic soil decreased the combined net N2O flux by up to 200% of the expected, combined net production rate from separate horizons. There was decoupling between gene abundances and biogeochemical outcomes.

Generalization of the results was done and may be enough from the potentials made on anaerobic conditions and with added NO3-. Possibly name would already indicate, that this was a laboratory experiment.

MS is well written and figures are clear. Methodology is mostly the same as used earlier by Billings and Tiemann, 2014, except that  $\delta$  15N is measured from N2O but not from produced N2 gas. There is some points needing further clarification.

Results are expressed as ng N2O-N g-1 h-1. It is unclear to me is this dry weight or fresh weight and same used in all soil weight (gen copies, added 15N etc.) based measurements? Soil samples in half-pint jars (240 ml) were 40 g mineral and 50 g of organic soil, and in combined experiment 20 g for mineral and 25 g for organic soil. Their bulk density (supposedly dry BD?) is  $\sim$ 0.1 for organic and  $\sim$ 0.7 g cm-3 for mineral soil. is there wet bulk density also available in order to compare thinks based on volume of the soil. This may explain why actual soil volumes in jars are different, it is not explained further. Or how close volumes are, when in both WHC is adjusted to 80% (which is of course a big difference in water content). Knowing actual volumes is especially important in combined setting, where volumes probably have an effect to ratios of produced and consumed N2O. Also, in methods (r: 252) the 0.25 g added soil (ww,fresh weight, dw) for functional gene analyses is unclear, and was this wet weight of soil straight from incubation flasks, and thus about 80% WHC? And the result based on this added amount of fresh (WHC 80%) or dry weight of added soil. It may also be worth to mention this and possible volume differences in discussion regarding functional gene abundances. I have difficulties to understand

SOILD
this tracing method (could be also my lack of knowledge), so maybe you explain it a bit more carefully. Why not adding 15N-N2O in the beginning and measuring 15N2 would not work? When you add 15N-NO3-, you assume that it will first produce enough big amount of 15N-N2O in three initial hours. From this concentration increases in 15N-N2O (production > consumption) or decreases (production

---

## Referee Comment (RC3) · Anonymous Referee #3 · 7 May 2020

In this manuscript the authors present a laboratory-based study to address three uncertainties in climate projections of net N2O fluxes from soil: (1) short v long-term responses to warming, (2) interactions among soil horizons, and (3) temperatures responses of different steps in the denitrification pathway. While the study itself is sound (although see my comment below about clarification of how net N2O fluxes were estimated), the authors treat denitrifiers superficially and interpret their results without deep consideration of the mechanisms driving the observed patterns—that is, they never mention any of the known controls on nitrate reduction and nitrous oxide reduction by denitrifiers and how warming or soil property differences among soil horizons would affect those controls. Below I detail some of my concerns and hope that my suggestions will help the authors improve the manuscript such that their findings can clearly advance our understanding of how warming affects soil denitrification.

The physiological rationale for enhanced rates of complete denitrification under long-term temperature regimes should be explained in order to justify this hypothesis. The following language is currently used to justify and describe this hypothesis: "less effective processing" by denitrifiers (Line 75) leading to more incomplete denitrification; denitrifying communities as "efficient transformers of NO3- to N2" (Line 137); "a soil denitrifying community well-adapted to its temperature regime is adept at complete denitrification"; and "denitrifier performance" (Line 415). But what is meant by "effective, "efficient," "adept," and "performance" as it relates to microbial physiology? Denitrifiers are mostly facultative anaerobes that can utilize various metabolisms other than nitrate reduction or nitrous oxide reduction depending on environmental conditions. There was no mention of controls on the actual processes of nitrate reduction and nitrous oxide reduction (e.g., nitrate availability, soil redox) anywhere in the manuscript, which severely undercuts the hypothesis and the interpretation of the results.

Lines 138-146: These three hypotheses are actually predictions (i.e., expected results). In the introduction, there is little justification presented for why these results would be expected other than similar patterns have been observed for heterotrophic respiration. As the authors acknowledge, denitrification is a more complicated process because it includes multiple enzymatic steps. But physiologically, the controls on denitrification are also different from heterotrophic respiration, and that needs to be considered.

The calculation of net N2O fluxes is core to the validity of the results of this study, so this vague statement on lines 202-203 needs to be clarified. Please explain what is meant by "the robustness of the final 60 h time point measure," how the multiple times points were used to verify this, and what is meant by "the net results of these samples." In addition, I recommend that the authors add a supplementary figure that shows the

net N2O fluxes calculated for each of the time points so that the readers can see what patterns got washed out by averaging the fluxes observed over the 60 hour incubation (Lines 219-222).

Line 227-242: Based on the equation presented in line 238, it is not the change in 15N enrichment of the N2O that is used to estimate N2O reduction rates as stated on line 227 but rather 15N2O abundance. Also, on lines 231, 234, 440, and elsewhere in the manuscript, only "15N2O" is referred to but it would be clearer to the reader if "15N2O abundance" was specified.

Lines 255-258: Please clarify what is meant by nirS and nosZ clade I as being "the most tractable indicators of N2O production and reduction." What does "tractable" mean, and how was this assessed? Please also specify which other functional gene primers were tested (including citations for the primers used) so that readers can interpret why these genes may have failed to amplify. Primers vary in their coverage of the diversity of microbes harboring a given functional gene, so the selected primers may not have been able to detect the relevant organisms present (see Ma et al. 2019, Environmental Microbiology). For example, only recently was a suite of primers developed that provides better coverage for the many subclades of nosZ clade II (see Chee-Sanford et al. 2020, Journal of Microbiological Methods).

Lines 411-413: This discussion of differences among the sites representing different long-term climate regimes needs to be expanded on. How would differences in soil organic matter input and nitrogen availability among the sites confound the interpretation of warming effects on N2O dynamics? The authors should consider how these potential confounding factors influence nitrate reduction and nitrous oxide reduction rates based on our understanding of controls on denitrification.

Lines 498-509: I would delete this paragraph about "contrasting efficiencies of N2O scavenging" which is speculative and not founded in an understanding of microbial physiology. What is meant by "efficiency"? The rationale presented does not consider

why microbes would reduce N2O nor differing conditions in mineral versus organic soils that would cause differences in N2O reduction rates in these two horizons.

---

## Author Response (AR1)

SOIL-2019-58

Topical Editor Comments and Response

07/07/2020

Dear Prof Sleutel,

We thank you for your supportive review and positive decision. We have addressed your questions and comments below (all highlighted line numbers refer to the revised pdf file), and particularly appreciate your helpful insights about the role of mineral soil N2O reduction. Together with the revisions in response to the three reviewers, this manuscript is much improved and we are very pleased with your support for publication in SOIL.

Sincerely,

Kate Buckeridge, on behalf of all co-authors

**Topical Editor Decision: Reconsider after major revisions (01 Jul 2020) by Steven Sleutel**
Comments to the Author:
The present research tried to tackle three relevant questions about global warming and denitrification at once. Not all was answered here in spite of the interesting experimental design but the authors did manage to bring up some relevant new questions. Particularly intriguing to see that mineral soil would be able to reduce N2O emitted from the organic soil layer, in spite of lower nosZ abundances. The introduction sets the stage well and makes a case for the presented work. I agree with the authors in that the three hypotheses are sufficiently well under-built and no further elaboration is needed. In response to comments raised by referees some further methodological clarifications are now presented as well. The results were already efficiently presented.

The discussion equally reads well, but I have a few comments:
While the methodology is relatively well thought through but it is somewhat regretful that incubations were carried out in a completely O2-free atmosphere, which really renders the results less representative. This needs to be brought up in the discussion near L434: N2O reduction was probably strongly promoted in these experiments with nearly no O2 present. In a field situation it may well be that N2O-reduction in the quite porous mineral layer is a much lesser likely process and so would then the 'soil horizon interactions' be a lot smaller.

>> We have added the following sentence to the end of the paragraph (prev l. 434, now l. 461-464) to clarify this possible experimental artifact: "A caveat to this soil horizon interaction is that while our $O_2$-limited experimental environment was necessary to promote denitrification, this design may have exaggerated bulk soil reduction processes that occur naturally in anaerobic microsites."

I partially do agree with referee 3 that generally observations on N2O emission and reduction are not very much discussed mechanistically, viz. with respect to drivers of these processes. But it is equally understandable that the authors chose not to do so as these experiments were run in O2 free air. One occasion where I hope the authors could still complement is L450: just why in your view would N2O reduction be particularly favoured in the mineral topsoil vs. organic layer? Perhaps because with a smaller porosity and probably also finer pore size distribution vs. the organic layer hot-spots for complete denitrification have always been more common in the mineral soil. This would then have led to a microbial community more fit (efficient) to reduce N2O? Or perhaps, N2O reduction is mostly simply uncommon in the organic layer as it is too porous and more directly in contact with the atmosphere so that N2O residence times are just too short to allow further reduction. I did find the hypothesis on L138-142 plausible but again, at present it is not discussed why the opposite result was found. It is for referees or readers not possible to reflect a bit on this with very little information provided on the sampled soil horizons. To the least also soil texture and SOC concentration should be added to table 1, but perhaps the authors also still see some room to further contemplate on just why mineral soil would harbour a microbial community more fit for N2O reduction.

>> We have added the following sentence to the end of the paragraph (prev l. 450, now l. 479-484) to clarify that this finding was in contrast with our original hypothesis, and a possible mechanism: "Contrary to our original hypothesis, shallow mineral soils in situ may be better suited than organic soils to $N_2O$ reduction, given that mineral soils experience frequent inputs of leached $NO_3^-$ and DOC from the surface organic soils, and represent a sudden change in the soil structure and porosity towards well-packed fines and smaller pores. These conditions may promote leachate pooling, anaerobic microsites, and a microbial community that proves more effective at reduction."

I recommend publication after final minor revision.

Some smaller comment:
L59 I suggest you omit "the emergent"

>> We have removed this phrase so that this sentence now reads: '…the temperature sensitivity of denitrification' (l. 62)
L70-79 should be shortened a bit. The message is clear but lengthy

>> We have done our best to make this more concise, but carefully, as the clarity of this concept has been a sticking point for several reviewers (l. 73-81)
L131 replace "predicted" by "expect" or "hypothesize"

>> We have replaced 'predicted' with 'expected' (l. 133)

L779 at the end of a 60 h incubation +

>> We are unsure of the meaning of this comment. The incubation was indeed conducted over 60 h.

Figure captions: in my view no need to repeat the three names of the sites once more, "coolest" etc. suffices

>> If at all possible, we would like to keep the site names in the figure captions, given that they are part of a long-standing latitudinal transect supporting many studies; multiple readers will be familiar with the sites.

L794 C instead of (B); In tab A the orange dot at 25°C is invisible, is that correct?

>> The orange dot is behind the red dot – both sets of error bars are evident, and now that we have changed the shapes, the orange square is now visible behind the red triangle.

L400 I wonder if "our predictions" is really the best term here. You are mainly bringing up an expectation/hypothesis, no?

>> We have replaced 'predictions' with 'hypothesis' (l. 421)

L425 'increased net N2O production to temperature' sounds awkward

>> We have replaced 'to warming' with 'with higher temperatures' (l. 451)

L426 'did not contradict' seems a bit overly careful, should the authors want to then this phrasing could be omitted.

>> We removed 'did not contradict' (l. 452)

L442-444 is not that readable; the phrasing "our horizon interactions" should be reworded. Probably this sentence can do without "although these results do not contradict the possibility of mineral soil reduction"

>> We have rewritten this sentence as follows: 'Large variation in $^{15}N_2O$ abundance among forest sites led to no significant difference between soil horizons and did not allow us to confirm the direction of horizon interactions.' (l. 472-474)

SOIL-2019-58

Response to Reviewers

22/05/2020

Dear Reviewers,

We thank you for taking the time to review our manuscript and hope that we have addressed your questions clearly. Our answers follow your questions below and are preceded by '>>'. The line numbers refer to our post-review, revised version of the text.

Anonymous Referee #1

Overall, this is a very interesting manuscript that is presented in a clear and easy to follow manner. The objectives and hypotheses are presented well and are followed with an exceptional design and conclusions. I thought the use of the combined incubations was a novel approach to provide a link to landscape-scale processes from the experimental setup. The site selections provide a very useful […]. The manuscript presents very interesting results especially regarding the potential for N2O reduction in mineral soils, but the impact of this finding is somewhat muddled in the presentation of the data.

In figure 4, the combination effect is presented and presented as a percent of the 'expected N2O production rate', but the definition of this rate is not clear. The logic behind these calculations is not clearly explained in the manuscript itself.

Is the expected rate actually the rate at the end of the 60h incubation of the soils incubated in isolation?

>> We have added the following text to the methods (l. 251-259):

"To assess the potential for N2O to be reduced to N2 by denitrifiers in the other horizon when incubated together, we calculated the combination effect (ng N2O-N g dw-1 h-1) as the difference between observed net N2O fluxes when soil horizons shared the incubation headspace (observed) and the expected flux determined as the linear, additive effect of rate for horizons in separate headspaces (((organic + mineral)/2) = expected).  The combination effect was also expressed as a percent of the expected flux:

Combination effect (%)=  (observed - expected)/expected*100, where a negative combination effect implies reduction caused by inclusion of one of the horizons."

We have also added a condensed version of this text to the legend of Figure 4, to clarify how we calculated this combination effect.

The expected rate is from the soils incubated in isolation, but it is calculated the same way as net rate, as the average of each time step.

So that the values expressed in Fig 4B are the absolute rates of Fig4A divided by the rates in Fig3 A &C?

>> The values in 4B are expressed as a percent of the expected, so they are calculated as ((observed - expected)/expected)*100

It would be important for the reader to understand the logic behind figure 4 and clearly explain the calculations within the manuscript.

>> As noted above, we hope these calculations are now clear.

Additionally, the results from the incubations of mineral soil horizons demonstrate low rates of N2O production, but without confirmation of 15N-N2 measurements, how are the authors confident that these low production rates correlate to high N2O reduction rates?

This can be addressed with the rate of 15N-NO3 throughout the experiment, but it is not clear in the text as the manuscript is currently written, please elucidate on this in the discussion.

>> We have elaborated on our explanation of the two methods of deducing that mineral soils are the dominant reducers in the discussion, supported primarily from the lower net N2O flux in our combination incubation (lower than organic soil alone), and we show that our 15N2O method does not confirm or deny the evidence for mineral soil reduction of organic horizon efflux (l. 451-474). We have also added an additional explanation when the isotopic results are first presented (l. 348-354).

Finally, one very minor comment regarding the figures. To help distinguish between the incubated soils, please use symbols additional to the colors.

>> We have remade the figures to include treatment-specific shapes.

Anonymous Referee #2

The study is important, since emissions from terrestrial systems dominate N2O fluxes which may be further enhanced by warming climate leading also to warming of the soil. In this study, the key studied function was denitrification, both producing and consuming N2O. The N2O flux potentials were measured from boreal ecosystem soils where mean annual temperature spans from 0 – 5.2 C. Furthermore, incubations by adding 15N-KNO3 were done in temperature range from 5 to 25 C in order to see the effect of warming to fluxes. Both short-term warming and a long-term warmer climate enhance net N2O production, and N2O production was bigger than N2O reduction during the incubation. There was reduction of N2O by mineral soil, not by organic soil. Combining horizons of mineral soil and organic soil decreased the combined net N2O flux by up to 200% of the expected, combined net production rate from separate horizons. There was decoupling between gene abundances and biogeochemical outcomes. Generalization of the results was done and may be enough from the potentials made on anaerobic conditions and with added NO3-. Possibly name would already indicate, that this was a laboratory experiment. MS is well written and figures are clear. Methodology is mostly the same as used earlier by Billings and Tiemann, 2014, except that _ 15N is measured from N2O but not from produced N2 gas.

There is some points needing further clarification.

Results are expressed as ng N2O-N g-1 h-1. It is unclear to me is this dry weight or fresh weight and same used in all soil weight (gen copies, added 15N etc.) based measurements?

>> All results are presented corrected by the soil oven dry weight. We have now clarified that in the text (l. 222-3, 248, 290).

Soil samples in half-pint jars (240 ml) were 40 g mineral and 50 g of organic soil, and in combined experiment 20 g for mineral and 25 g for organic soil. Their bulk density (supposedly dry BD? >> Yes) is _0.1 for organic and _0.7 g cm-3 for mineral soil. is there wet bulk density also available in order to compare thinks based on volume of the soil.

>> Bulk density at 80% WHC was not determined, as bulk density was calculated from field conditions, prior to the incubation.

This may explain why actual soil volumes in jars are different, it is not explained further. Or how close volumes are, when in both WHC is adjusted to 80% (which is of course a big difference in water content). Knowing actual volumes is especially important in combined setting, where volumes probably have an effect to ratios of produced and consumed N2O.

>> The organic:mineral soil mass ratio were chosen to represent (approximately) the average ratio of organic:mineral soil mass ratio in the peds that were collected from all the forest sites. The dry mass and fresh volumes of the two soil horizons were not similar; there was less dry mass and more fresh volume occupied by the less-dense organic soil. However, in spite of this attempt to mimic realistic O and mineral horizon ratios, organic and mineral soil were next to each other, as opposed to incubating the organic layer on top of the mineral soil. Thus, O horizons and mineral soils shared the same headspace, as described in the methods (l. 186-190). Regardless of our attempts to mimic (in part) some relationships between organic and mineral soils in situ, we recognize that our incubation is not representative of ecosystem-level in situ process rates, which is why all data are reported on a mass, rather than an areal, basis.

Also, in methods (r: 252) the 0.25 g added soil (ww,fresh weight, dw) for functional gene analyses is unclear, and was this wet weight of soil straight from incubation flasks, and thus about 80% WHC? And the result based on this added amount of fresh (WHC 80%) or dry weight of added soil. It may also be worth to mention this and possible volume differences in discussion regarding functional gene abundances.

>> Actual mass extracted was approximately 0.25 g fresh (80% WHC) weight, as per extraction kit protocol – we have updated the text to indicate this (l. 269). The precise amount extracted was recorded for each sample and converted to oven dry weight equivalent based on the dry:fresh weight ratio of an oven-dried subsample collected at the same time (post-incubation). All gene abundance data (and all data) are therefore presented as corrected by soil dry weight, as now indicated (l. 289-90).

I have difficulties to understand this tracing method (could be also my lack of knowledge), so maybe you explain it a bit more carefully (please see next answer). Why not adding 15N-N2O in the beginning and measuring 15N2 would not work?

>> Adding 15N-N2O and tracing it to 15N2 would theoretically allow us to detect N2O consumption, but it would require quantification of the dilution rate of 15N-N2O with 14N-N2O to quantify N2O production. In theory, this could work, but the applied approach of watching 15N-labeled NO3- be transformed into N2O is also robust, and is the protocol used in our lab, successfully to date (Billings and Tiemann 2014).

When you add 15N-NO3-, you assume that it will first produce enough big amount of 15N-N2O in three initial hours. From this concentration increases in 15NN2O (production > consumption) or decreases (production < consumption) are visible in the incubation time 60h. In this method you also assume, that 14N N2O released from soil own N stores is not diluting 15N + 14N -N2O.

In equation at row 238, it would be easier to reader to show the real times (initial 3 h and final 60 h).

We do not assume that all 15NO3 will be consumed in the first three hours, rather that the rate of 15N2O production will decline as substrate declines, whereas consumption will increase relative to production as we approach 60 h, a reasonable assumption given that NO3 pools decline over the course of the incubation and the substrate for consumption, N2O, increases over this same time frame (l. 236-250). We do assume that 14N2O will not be an important agent of diluting the isotopically labeled N2O pool, because the tracer was highly concentrated (l. 234-236).

Also jump from added _15N 3000 ‰ first to ng 15N- N2O g-1 soil (dw,ww, fresh?) maybe needs to be explained more clearer. It would be nice to see (or have a reference) how you get from headspace N2O ppm:s and _15N values to ng g-1 15N-N2O in headspace, since there is also lot of 14N-N2O added. In any case recovery of 15N as N2O is big, almost 75% at the warmest experiment at the time point 60 h.

>> We have hopefully clarified these calculations by including calculation steps from ppm to rates on l. 220-222 and rates to 15N2O on l. 241-244.

Would be nice to see also course of N2O concentration increase with time – and d15N-N2O for the time points used. Maybe as supplement.

>> We have added net N2O concentration over time as Supplementary Figure 1. 15N2O was only analysed at 3h and 60h.

Some typos (or not) and just asking:

r, 226 : there is range of ppm, but what was the range of d15N-N2O standards, 0.1 ‰ precision looks for me extremely good in highly enriched N2O.

>> IRMS precision is typically presented as the standard deviation across five natural abundance standards, and 0.1 ‰ precision (standard deviation) is the value that UC Davis SIF provided to us. We have clarified this in the text (l. 230-1). We agree that it is normal and likely that enriched samples will have a lower precision (higher standard deviation), although we did not submit five replicate enriched samples for comparative analysis.

r. 347 and elsewhere. " g-1 _" between standard deviation (or error?) or after that.

>> We agree that g-1 should be after the standard error. We have changed this for all the gene abundances in this section (l. 360-377).

r. 246, r. 252: is 0.25 g also based on fresh weights?

>> Yes, we have clarified this in the text (l. 269).

reference list: many typos in subscripts N2O, spaces N 2 O and letters, like "Dur??n"

>> Thank you for checking these, we believe we have corrected all the typos in the references.

Good luck with MS!

>> Thank you!

Anonymous Referee #3

In this manuscript the authors present a laboratory-based study to address three uncertainties in climate projections of net N2O fluxes from soil: (1) short v long-term responses to warming, (2) interactions among soil horizons, and (3) temperatures responses of different steps in the denitrification pathway. While the study itself is sound (although see my comment below about clarification of how net N2O fluxes were estimated), the authors treat denitrifiers superficially and interpret their results without deep consideration of the mechanisms driving the observed patterns that is, they never mention any of the known controls on nitrate reduction and nitrous oxide reduction by denitrifiers and how warming or soil property differences among soil horizons would affect those controls. Below I detail some of my concerns and hope that my suggestions will help the authors improve the manuscript such that their findings can clearly advance our understanding of how warming affects soil denitrification.

The physiological rationale for enhanced rates of complete denitrification under longterm temperature regimes should be explained in order to justify this hypothesis. The following language is currently used to justify and describe this hypothesis: "less effective processing" by denitrifiers (Line 75) leading to more incomplete denitrification; denitrifying communities as "efficient transformers of NO3- to N2" (Line 137); "a soil denitrifying community well-adapted to its temperature regime is adept at complete denitrification"; and "denitrifier performance" (Line 415). But what is meant by "effective, "efficient," "adept," and "performance" as it relates to microbial physiology?

>> We agree that these terms were not well-explained. We have changed the wording at these locations to use a controlled vocabulary. Instead of using this large number of terms, we use the term "effective" and define what we mean by it on lines 141-142. There, we state our interpretation of more 'effective' denitrification as more complete denitrification (i.e. higher N2:N2O ratio) (l. 76-77). This concept is valuable when considering long- and short-term responses of denitrifiers to environmental conditions.

Denitrifiers are mostly facultative anaerobes that can utilize various metabolisms other than nitrate reduction or nitrous oxide reduction depending on environmental conditions. There was no mention of controls on the actual processes of nitrate reduction and nitrous oxide reduction (e.g., nitrate availability, soil redox) anywhere in the manuscript, which severely undercuts the hypothesis and the interpretation of the results.

>> By working with organic matter-rich soils, we attempted to maximize natural C substrate availability and make that point on line 200. While we agree that the absence of O2 (and associated redox potential of a system) is a dominant driver of denitrification, one of the goals of the study was to investigate N2O production in upland, fairly well-aerated soils (l. 162-163).

We agree that the availability of C, NO3 and O2 dominate the effect of warming on the production of net N2O and have clarified this important point in the text by modifying some of our description of the study's aims (l. 58-61). Specifically, our aim is to explore issues related to temperature sensitivity beyond these proximate controls and their indirect effects. We expect that these proximate controls, or changes to them, will have indirect effects that are complexly packaged in our long-term warming treatment, and make this point in section 4.1 of the discussion (l. 406-410), and acknowledged the importance of these controls in the conclusion (l. 566-567).

Lines 138-146: These three hypotheses are actually predictions (i.e., expected results). In the introduction, there is little justification presented for why these results would be expected other than similar patterns have been observed for heterotrophic respiration. As the authors acknowledge, denitrification is a more complicated process because it includes multiple enzymatic steps. But physiologically, the controls on denitrification are also different from heterotrophic respiration, and that needs to be considered.

>> We respectfully disagree that there is little justification for our hypotheses. In the introduction, we have provided one paragraph explaining the rationale behind each of the 3 hypotheses, then summarized these justifications in the final prediction paragraph of the introduction. Each of these hypotheses addresses a knowledge gap, with varying levels of prior evidence. We have provided the knowledge gap, the evidence, and our evidence-based hypotheses, in addition to one prediction (short-term warming response), which is based on past results.

>> We agree with the reviewer that physiologically, the controls on denitrification and respiration are probably different because they are carried out by different microbes, or the same microbes in different environmental conditions. Nonetheless, we believe we are justified to use an example of a community-level, physiological (i.e., microbial respiration) response to warming and to hypothesize that it may also apply to denitrification. Because we have very little a priori knowledge about long-term denitrifier responses to warming, and because aerobic respiration and denitrification both represent microbial respiratory pathways, using results from studies of heterotrophic respiratory responses to warming to predict one result in this study seems a valid starting point for developing an analogous knowledge base about denitrifiers. Indeed, it tests the idea that warming can prompt respiratory pathways across microbial taxa to respond in similar ways, which represents a way forward for more generalized hypotheses about ecosystem responses to warming.

The calculation of net N2O fluxes is core to the validity of the results of this study, so this vague statement on lines 202-203 needs to be clarified.

Please explain what is meant by "the robustness of the final 60 h time point measure," how the multiple times points were used to verify this, and what is meant by "the net results of these samples."

>> We apologise that this was unclear. We have now removed that sentence since our method of averaging is already described in greater detail elsewhere (l. 222-226).

In addition, I recommend that the authors add a supplementary figure that shows the net N2O fluxes calculated for each of the time points so that the readers can see what patterns got washed out by averaging the fluxes observed over the 60 hour incubation (Lines 219-222).

>> We have added this figure as Supplementary Figure 1.

Line 227-242: Based on the equation presented in line 238, it is not the change in 15N enrichment of the N2O that is used to estimate N2O reduction rates as stated on line 227 but rather 15N2O abundance.

Also, on lines 231, 234, 440, and elsewhere in the manuscript, only "15N2O" is referred to but it would be clearer to the reader if "15N2O abundance" was specified.

>> Thank you, we have changed this as suggested.

Lines 255-258: Please clarify what is meant by nirS and nosZ clade I as being "the most tractable indicators of N2O production and reduction." What does "tractable" mean, and how was this assessed?

>> We describe this as 'based on successful amplification of these genes across all samples' (l. 277).

Please also specify which other functional gene primers were tested (including citations for the primers used) so that readers can interpret why these genes may have failed to amplify.

>> We have updated this information in the text (l. 273-275) and added the table below to supplementary data.

| Gene | Primer | Sequence (5' to 3') | Reference |
|---|---|---|---|
| Nitrite reductase (NO$_2^-$ to NO) | nirK-876F | ATY GGC GGV AYG GCG A | Henry et al. 2004 |
| nirK | nirK-1040R | GCC TCG ATC AGR TTR TGG TT | |
| Nitrite reductase (NO$_2^-$ to NO) | **nirS-cd3aF** | GTS AAC GTS AAG GAR ACS GG | Throbäck et al., 2004 |
| nirS | **nirS-R3cd** | GAS TTC GGR TGS GTC TTG | |
| Nitric oxide reductase (NO to N$_2$O) | cnorB2F | GAC AAG NNN TAC TGG TGG T | Braker and Tiedje, 2003 |
| norB | cnorB7R | TGN CCR TGN GCN GCN GT | |
| Nitric oxide reductase (NO to N$_2$O) | **nosZ-F** | CGY TGT TCM TCG ACA GCC AG | Röche et al., 2002 |
| nosZ | **nosZ-R** | CAT GTG CAG NGC RTG GCA GAA | |
| Nitrous oxide reductase (N$_2$O to N$_2$) | nosZ-II-F | CTI GGI CCI YTK CAY AC | Jones et al., 2013 |
| nosZ II | nosZ-II-R | GCI GAR CAR AAI TCB GTR C | |

Primers vary in their coverage of the diversity of microbes harboring a given functional gene, so the selected primers may not have been able to detect the relevant organisms present (see Ma et al. 2019, Environmental Microbiology).

For example, only recently was a suite of primers developed that provides better coverage for the many subclades of nosZ clade II (see Chee-Sanford et al. 2020, Journal of Microbiological Methods).

>> We agree that this may have contributed to our lack of success with these primers. We do not delve into this discussion because we did not spend extensive lab time optimizing PCR conditions for all primers, and because it was always our intent to choose representative (as opposed to exhaustive) denitrifier production and consumption genes.

Lines 411-413: This discussion of differences among the sites representing different long-term climate regimes needs to be expanded on. How would differences in soil organic matter input and nitrogen availability among the sites confound the interpretation of warming effects on N2O dynamics? The authors should consider how these potential confounding factors influence nitrate reduction and nitrous oxide reduction rates based on our understanding of controls on denitrification.

>> We have clarified in the text that we think these historical differences may be directly important controls for N2O production in situ, but less so in our incubation where we added a small pulse of NO3. Therefore, it is more likely that differences in regional microbial responses to short-term warming reflect community-level microbial acclimation to their historical conditions (l. 434-439).

Lines 498-509: I would delete this paragraph about "contrasting efficiencies of N2O scavenging" which is speculative and not founded in an understanding of microbial physiology. What is meant by "efficiency"? The rationale presented does not consider why microbes would reduce N2O nor differing conditions in mineral versus organic soils that would cause differences in N2O reduction rates in these two horizons.

>> We respectfully disagree that contrasting enzyme efficiencies are not based in an understanding of microbial physiology, although we do agree that we know very little about how environmental conditions impact the enzyme efficiencies for the multiple steps in the denitrification pathway. We have added a sentence to illustrate that community structure and resource availability (conditions which differ between soil horizons) have been demonstrated to alter enzyme soil microbial efficiency (l. 533-536). We have also elaborated on the discussion introduced in the previous paragraph about the need to assay more enzyme genes, and suggest that further research into nosZ clade II may help explain the apparent decoupling between rates and gene abundances between organic and mineral soil found in this study (l. 540-543).

[revised manuscript text omitted]

---

## Author Response (AR2)

SOIL-2019-58
Executive and Topical Editor Decision and Response
07/15/2020

Dear Profs Sleutel and Six,

We thank you again for your supportive reviews and positive decision, we are very pleased to be published in SOIL and are excited about the newly announced impact factor!
We have numbered our two equations and replaced all range short dashes (-) with and en dash (–). We have also included our data/code availability statement with a link to a Zenodo DOI.

Sincerely,
Kate Buckeridge, on behalf of all co-authors

**Executive Editor Decision: Publish subject to technical corrections** (14 Jul 2020) by Johan Six
Comments to the Author: Dear Authors,

First of all, thank you for chosing SOIL as an outlet for your work! Also, thanks for working with the reviewers and the topical editor to improve the manuscript. It is with great pleasure that I can accept the manuscript after some further minor revisions as suggested by the Topical editor:

1° All equations should be numbered sequentially with Arabic numerals in parentheses on the right-hand side, e.g. (1), (2). If too long, split them accordingly. When using Word, the equation editor and not the graphic mode should be used under all circumstances. They should also be referred to in the text by the abbreviation "Eq." and the respective number in parentheses, e.g. "Eq. (14)". However, when the reference comes at the beginning of a sentence, the unabbreviated word "Equation" should be used, e.g.: "Equation (14) is very important for the results; however, Eq. (15) makes it clear that..."

2° En dashes (–) are used to indicate, among other things, ranges (e.g. 12–20 months). -> check L172, 176, 230,

Best regards,
Jo

**Topical Editor Decision: Publish subject to technical corrections** (14 Jul 2020) by Steven Sleutel
Comments to the Author:
All previously criticism raised has been appropriately addressed. The manuscript is now nearly ready for publication in SOIL. Only a two minor technical issues remain to be resolved.

1° All equations should be numbered sequentially with Arabic numerals in parentheses on the right-hand side, e.g. (1), (2). If too long, split them accordingly. When using Word, the equation editor and not the graphic mode should be used under all circumstances. They should also be referred to in the text by the abbreviation "Eq." and the respective number in parentheses, e.g. "Eq. (14)". However, when the reference comes at the beginning of a sentence, the unabbreviated word "Equation" should be used, e.g.: "Equation (14) is very important for the results; however, Eq. (15) makes it clear that..."

2° En dashes (–) are used to indicate, among other things, ranges (e.g. 12–20 months). -> check L172, 176, 230,